# Temporary Reversal of Hepatoenteric Collaterals during $^{90}$Y Radioembolization Planning and Administration

Peiman Habibollahi [1], Bruno C. Odisio [1], Varshana Gurusamy [2], Joshua D. Kuban [1], Rony Avritscher [1], Mohamed E. Abdelsalam [1], Beth A. Chasen [3], Ravi Murthy [1] and Armeen Mahvash [1,*]

1   Department of Interventional Radiology, The University of Texas MD Anderson Cancer Center, Houston, TX 77030, USA
2   Department of Radiology, Medical University of South Carolina, Charleston, SC 29208, USA
3   Department of Nuclear Medicine, The University of Texas MD Anderson Cancer Center, Houston, TX 77030, USA
*   Correspondence: armeen.mahvash@mdanderson.org; Tel.: +713-563-7340

**Abstract:** Purpose: This paper aims to evaluate the safety and efficacy of the temporary redirection of blood flow of hepatoenteric collaterals using a balloon catheter in the common hepatic artery (CHA) to prevent the nontarget deposition of $^{90}$Y microspheres. Materials and Methods: In this retrospective single-center study of patients who received $^{90}$Y radioembolization (RE) from September 2010 to September 2015, diagnostic (67 patients) or treatment (72 patients) angiograms with the attempted use of a balloon catheter in the CHA to temporarily direct blood flow away from the hepatoenteric arteries were analyzed. SPECT/CT nuclear scintigraphy was performed after both diagnosis and treatment. Results: Overall, only 12 hepatoenteric arteries in 11 patients required embolization due to persistent hepatoenteric flow despite the use of the balloon occlusion technique in a total of 86 patients. Physicians performed the $^{90}$Y RE using balloon occlusion with glass ($n = 22$) or resin ($n = 50$) microspheres. Over 80% administration of the prescribed $^{90}$Y dose was accomplished in 34 (67%) resin and 20 (95%) glass microsphere patients. Post-treatment $^{90}$Y RE scintigraphy confirmed the absence of extrahepatic activity in all patients. One grade 2 gastrointestinal ulcer was present after 90 days of follow-up. Conclusion: Temporary CHA occlusion with a balloon catheter is a reliable and reproducible alternative to the conventional coil embolization of hepatoenteric arteries during diagnostic Tc-99m macroaggregated albumin and therapeutic $^{90}$Y RE delivery.

**Keywords:** radioembolization; balloon occlusion; hepatoenteric collaterals

## 1. Introduction

Radioembolization (RE) or selective internal radiation therapy with yttrium-90 ($^{90}$Y) resin or glass microspheres is a locoregional treatment option with clinical benefit for patients suffering from unresectable primary or secondary liver malignancies [1–4]. This technique yields the selective delivery of radioactive microspheres into the arteries supplying a tumor to maximize the radiation dose to it and to achieve a response while minimizing the radiation dose to normal liver tissue by taking advantage of differential blood flow to normal and tumoral tissue to avoid radiation-induced liver disease.

During RE, radioactive microspheres are passively implanted into the tumor by the hepatopetal blood flow [3]. Thus, detailed mapping of the hepatic arterial anatomy before administration of the dose is essential to ensure adequate treatment and minimize nontarget RE. A nontarget delivery of $^{90}$Y microspheres to the gastrointestinal (GI) tract is a complication of RE, resulting in GI symptoms and ulcers [5]. The rate of GI complications after $^{90}$Y RE in prior studies varies widely, but irrespective of the rate, RE-induced GI ulcers may be resistant to medical treatment and require surgical resection [6,7]. Therefore, every effort should be made to minimize the risk of the nontarget delivery of microspheres [3].

The common practice used to minimize the nontarget delivery of microspheres is to embolize hepatoenteric vessels, including the right gastric artery (RGA) and gastroduodenal artery (GDA), depending on the planned catheter position for infusion. However, this increases the treatment time and cost. The transient reversal of blood flow within hepatoenteric blood vessels by creating a temporary occlusion in the common hepatic artery (CHA) using a balloon was first reported by Nakamura et al. [8] in 1985. It was later used in a small number of patients for the administration of glass $^{90}$Y microspheres (TheraSpheres; Boston Scientific, Marlborough, MA, USA) by Andrews et al. [9] and resin $^{90}$Y microspheres (SIR-Spheres; Sirtex, Woburn, MA, USA) by Mahvash et al. [10]. However, most previous studies included a small number of patients. Herein, we report on the utility of creating hepatopetal flow within hepatoenteric arteries using the balloon occlusion technique for diagnostic and therapeutic RE in a large cohort of patients at our institution who underwent $^{90}$Y RE planning and delivery.

## 2. Materials and Methods

This single-center, retrospective, observational study was performed at a tertiary referral academic center. Institutional review board approval was obtained, which included a waiver of participant consent given the retrospective study design. The MD Anderson Department of Interventional Radiology pre-existing procedure log was reviewed, and all patients who underwent RE from September 2010 to September 2015 were identified. All RE procedures were screened for the utilization of the occlusion balloon, and patients who did not undergo balloon occlusion were excluded.

After identifying the study patients, their demographic and clinical characteristics were gathered by reviewing institutional electronic medical records. These characteristics included the primary cancer diagnosis, date of diagnosis, and history of prior liver surgery/resection. Data regarding the RE procedures performed were also collected, consisting of the following: date of the procedure; type of hepatic arterial anatomy per Michel's classification of hepatic arterial anatomy; name of the hepatoenteric arteries; number of hepatoenteric arteries; direction of flow within the hepatoenteric blood vessels before and after balloon occlusion on planning and treatment; arteries in need of embolization or arteries embolized due to operator preference; location of delivery of Tc-99m macroaggregated albumin (MAA); presence of extrahepatic uptake; type (lobar vs. whole liver) of and prescribed $^{90}$Y dose for RE; type of microspheres used (TheraSpheres vs. SIR-Spheres); treatment outcome (success or failure); potential reason for treatment failure; and adverse events.

All procedures were performed by board-certified interventional radiology faculty with more than 5 years of experience. Conscious sedation was administered by nursing staff supervised by the physician performing the procedure in the angiography suite. For the planning study, access to the femoral artery was obtained using real-time ultrasound guidance, and a vascular sheath was inserted into the artery. Selective arteriography of the superior mesenteric and celiac arteries was performed. Based on the findings, further diagnostic angiography within the hepatic circulation was carried out using a combination of a wire and a microcatheter. The hepatoenteric vessels were identified using a combination of diagnostic arteriography, cone-beam CT, or in room dedicated contrast enhanced intra arterial CTs. When hepatoenteric blood vessels with extrahepatic blood flow were identified, which required embolization for the safe administration of $^{90}$Y microspheres, the balloon occlusion technique was used. A 4 or 5.5 Fr over the wire balloon occlusion catheter (Edwards Lifesciences, Santa Ana, CA, USA) was advanced into the CHA, and an angiogram was performed at a 1–3 mL/s injection rate of iodinated contrast to redemonstrate the hepatoenteral artery of interest. In a small initial cohort of patients, 4000 U of heparin was administered to prevent arterial thrombosis, the occlusion balloon was inflated, and repeat angiographies were performed using an injection rate of 1–3 mL/s of iodinated contrast to look for successful flow reversal by the absence of flow in the previously visualized arteries. If successful, Tc-99m MAA was delivered as planned through a coaxial microcatheter or from the balloon catheter in the desired position

with the balloon inflated. Post planning, single-photon emission computed tomography (SPECT)/computed tomography (CT) nuclear scintigraphy was then used to exclude the extrahepatic deposition of Tc-99m MAA.

After dose calculation, patients returned for the administration of $^{90}$Y microspheres. A technique similar to planning was used during administration by repeating the arteriogram through the occlusion balloon to visualize the hepatoenteric arteries and demonstrate the absence of flow and/or successful flow reversal after inflating the balloon. Flow reversal was confirmed via initial visualization of the hepatoenteric collaterals without balloon inflation and by the subsequent absence of extrahepatic flow when the balloon was inflated. For whole liver infusions, the microspheres were then administered through a coaxial microcatheter or 4 Fr balloon catheter at a previously determined position. A coaxial microcatheter was used for all selective lobar microsphere infusions. Figures 1–3 shows images from planning and treatment angiography, as well as from nuclear scintigraphy, using SPECT/CT from three select patients.

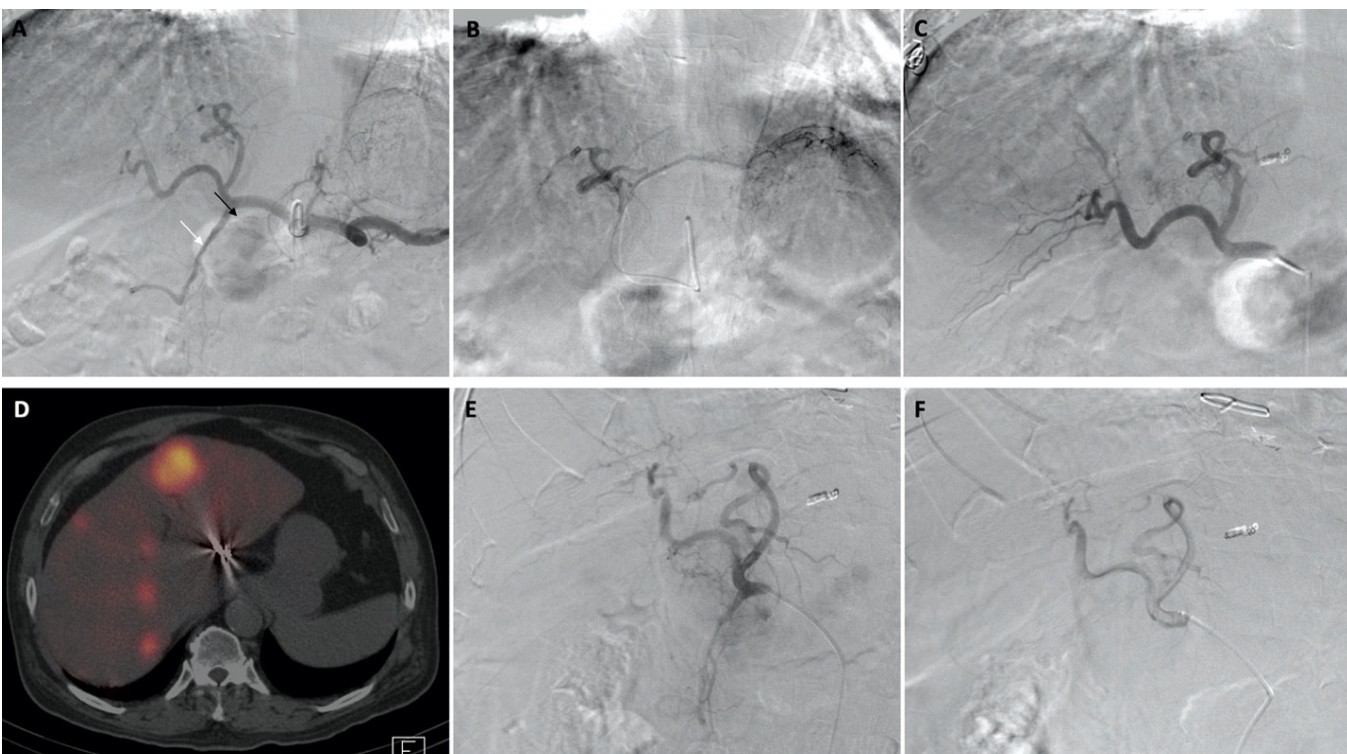

**Figure 1.** Images of a patient given diagnostic Tc-99m MAA injections and therapeutic $^{90}$Y RE. Digital subtraction angiography through the celiac axis showed hepatofugal flow within the GDA (white arrow) and RGA (black arrow) (**A**). Further, selective left hepatic arteriography showed an accessory LGA (**B**), which was embolized. A repeat celiac arteriogram with the balloon occlusion technique showed flow reversal in the RGA and GDA (**C**). Nuclear scintigraphy using SPECT/CT after Tc-99m MAA administration showed adequate distribution of Tc-99m MAA within the liver without significant extrahepatic uptake (**D**). A repeat celiac arteriogram through the balloon catheter during $^{90}$Y RE redemonstrated patency of the GDA and RGA (**E**) and flow reversal with inflation of the balloon (**F**).

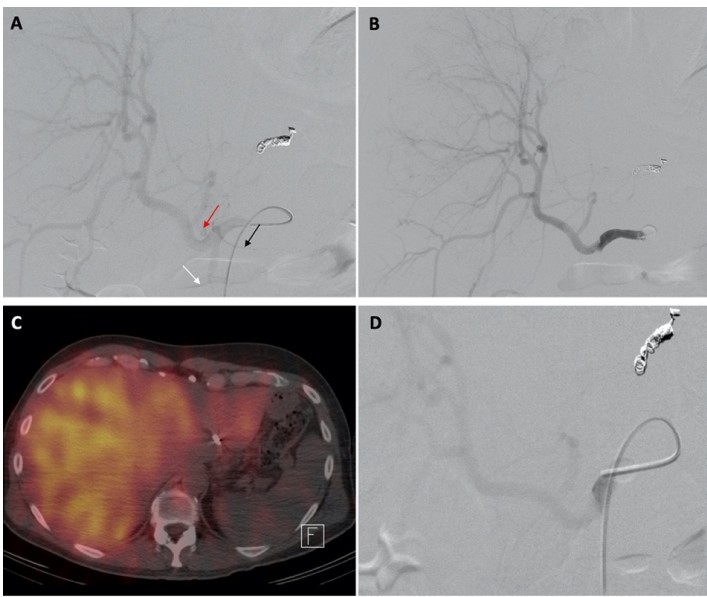

**Figure 2.** Images of a patient given diagnostic Tc-99m MAA injections and therapeutic ⁹⁰Y RE. Digital subtraction angiography from the celiac axis showed hepatofugal flow within the GDA (white arrow), RGA (black arrow), and supraduodenal artery (red arrow) (**A**). The accessory LHA was embolized for flow redistribution. A repeat celiac arteriogram with the balloon occlusion technique showed flow reversal in all three hepatoenteric arteries (**B**). Nuclear scintigraphy using SPECT/CT after Tc-99m MAA administration showed satisfactory distribution of it within the liver without significant extrahepatic uptake (**C**). A repeat celiac arteriogram through the inflated balloon catheter during ⁹⁰Y RE demonstrated flow reversal (**D**).

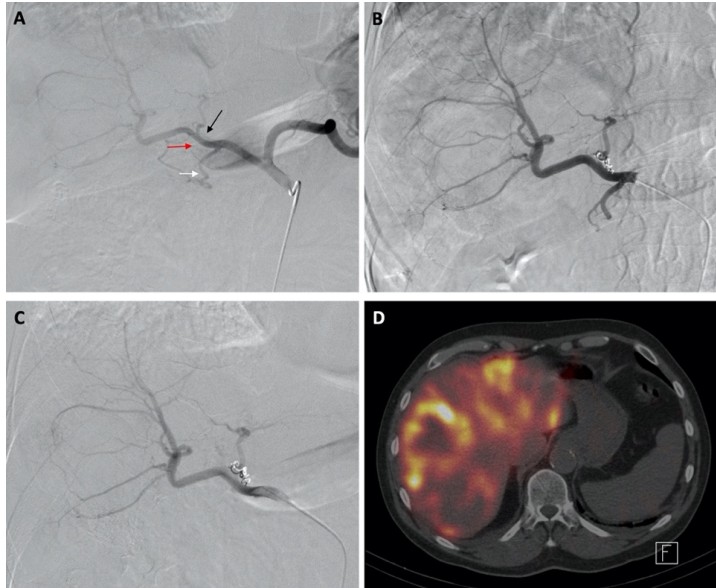

**Figure 3.** Images of a patient given diagnostic Tc-99m MAA injections. Digital subtraction angiography from the celiac axis showed hepatofugal flow within the GDA (white arrow), RGA coming from the LHA (black arrow), and retroportal artery (red arrow) (**A**). A repeat celiac arteriogram through the balloon catheter in the CHA after RGA embolization redemonstrated the hepatofugal flow in the GDA and retroportal artery (**B**) with successful flow reversal after inflating the balloon (**C**). Nuclear scintigraphy using SPECT/CT after Tc-99m MAA administration showed satisfactory distribution of it within the liver without significant extrahepatic uptake (**D**).

Patients were evaluated via telephone at 1 month after treatment and standard clinical evaluations were performed by the primary oncologist at 3-month intervals following treatment with repeat follow-up contrast-enhanced CT or magnetic resonance imaging. Adverse events were recorded using the National Cancer Institute Common Terminology Criteria for Adverse Events (version 5.0).

During the study period, 67 patients underwent diagnostic Tc-99m MAA injection and 77 underwent therapeutic $^{90}$Y RE using an occlusion balloon catheter in the CHA to temporarily reverse the blood flow from the hepatoenteric arteries toward the liver. Patient characteristics are listed in Table 1.

**Table 1.** Characteristics of the study patients.

| Characteristic | *n* (%) | |
| --- | --- | --- |
| | Diagnostic Tc-99m MAA Administration (*n* = 66) | Therapeutic $^{90}$Y RE Administration (*n* = 72) |
| Mean (±SD) age, years | 58.5 ± 15.1 | 57.9 ± 15.1 |
| Female (%) | 19 (29) | 25 (35) |
| Primary diagnosis | | |
| Hepatocellular carcinoma | 20 (30) | 22 (31) |
| Neuroendocrine tumor | 19 (29) | 25 (35) |
| Colorectal cancer | 18 (27) | 15 (21) |
| Carcinoid tumor | 4 (6) | 5 (1) |
| Medullary thyroid cancer | 1 (2) | 1 (1) |
| GI stromal tumor | 1 (2) | - |
| Adrenocortical carcinoma | 1 (2) | 1 (1) |
| Ameloblastoma | 1 (2) | 1 (1) |
| Small round cell tumor | 1 (2) | 1 (1) |
| Cholangiocarcinoma | - | 1 (1) |
| Michel's classification of hepatic arterial anatomy | | |
| Type 1 | 53 (80) | 56 (78) |
| Type 2 | 3 (5) | 1 (1) |
| Type 3 | 2 (3) | 4 (6) |
| Type 4 | 1 (2) | 1 (1) |
| Type 5 | 4 (6) | 6 (8) |
| Type 6 | - | 1 (1) |
| Type 9 | 2 (3) | 2 (3) |
| LHA from CHA | 1 (2) | 1 (1) |

SD: standard deviation; MAA: macro-aggregated albumin; RE: radioembolization; GI: gastrointestinal; LHA: left hepatic artery; CHA: common hepatic artery.

## 3. Results

During diagnostic Tc-99m MAA injection, balloon occlusion was successful in 66 patients (99%): in 1 patient, the physician abandoned the technique due to spasm of the CHA before balloon advancement, and the planning was completed using the standard technique. Four patients in this group underwent a previous surgical resection, consisting of a right hepatectomy, left hepatectomy, partial left hepatectomy and radiofrequency ablation, and wedge resection in one patient each. The visualized outcomes after balloon inflation are listed in Table 2. Reversal of the blood flow occurred in up to three hepatoenteric branches as shown in Table 2. There was successful flow reversal in 60/66 patients (90%) based on the angiography. A total of fourteen vessels underwent embolization prior to the administration of Tc-99m MAA. In particular, the purpose of the embolization was to redistribute the blood flow in five vessels (three accessory left hepatic arteries (LHAs), one replaced right hepatic artery (RHA), and one segment IV branch). Additionally, physicians performed embolization due to user preference and despite the successful flow reversal with balloon inflation in three RGAs. The RGA was the most common artery in need of embolization (4/66 [6%]), secondary to unsuccessful flow reversal, followed by the accessory left gastric artery (LGA; 2/66 [3%]).

Physicians delivered Tc-99m MAA in the CHA (*n* = 53), RHA (*n* = 2), LHA (*n* = 1), proper hepatic artery (*n* = 4), replaced RHA (*n* = 1), and separately in the RHA and LHA (*n* = 3), RHA and CHA (*n* = 1), replaced RHA and CHA (*n* = 1). Nuclear scintigraphy using SPECT/CT identified the extrahepatic uptake of Tc-99m MAA in eight patients, including two who had uptake in the stomach and proximal duodenum. Three patients had uptake in the pancreatic head due to missed small pancreatic branches supplying the head of the pancreas (two patients were treated with a split dose for the right and left hepatic lobes and one patient only received left lobar treatment), and three others had uptake in other locations, including the left hemidiaphragm from the left inferior phrenic artery coming off the celiac that did not reverse (patient was treated past this branch), porta hepatis from a small branch coming off the RHA (left hepatic lobe was treated), and falciform ligament from a branch of the LHA (patient was not treated secondary to major extrahepatic blood supply to tumor).

**Table 2.** Blood flow reversal in hepatoenteric arteries following balloon occlusion during diagnostic arteriography and planning.

| Arteries with Flow Reversal | Number of Patients with Flow Reversal (%) | Arteries without Flow Reversal | Coil Embolization | Extrahepatic Uptake on SPECT/CT |
|---|---|---|---|---|
| GDA only | 15 (23) | In two patients, the RGA was not visualized; in three patients, the RGA flow did not reverse after balloon inflation; in one patient, flow in both the RGA and retroduodenal artery did not reverse | Flow redistribution: accessory LHA in one patient and segment IV branch in one patient; operator preference: RGA in one patient; hepatoenteric embolization: RGA in three patients | Pancreatic uptake in two patients |
| RGA only | 2 (3) | In one of the patients, the GDA flow was reversed before balloon inflation; in the other patient, the GDA flow did not reverse | - | - |
| GDA and RGA | 34 (52) | - | Flow redistribution: accessory LHA in two patients and replaced RHA in one patient; hepatoenteric embolization: accessory LGA in two patients | Gastric uptake in one patient, pancreatic uptake in one patient, and porta hepatic uptake through a small RHA branch in one patient |
| GDA and supraduodenal artery | 2 (3) | In one patient, the RGA flow did not reverse after balloon inflation | Hepatoenteric embolization: the RGA was embolized in one patient | Falciform ligament uptake in one patient |
| GDA and retroportal artery | 1 (2) | - | Operator preference: the RGA was embolized | - |
| GDA, RGA, and supraduodenal artery | 8 (12) | In one patient, the left inferior phrenic artery was identified, and its blood flow did not reverse after balloon inflation | - | Duodenal uptake through a proximal GDA branch and paraumbilical uptake in one patient and left diaphragmatic and falciform ligament uptake in the patient without flow reversal in the left inferior phrenic artery |
| GDA, RGA, and cystic artery | 1 (2) | - | - | - |
| GDA, RGA, and retroduodenal artery | 1 (2) | - | - | - |
| GDA, RGA, and segment 4b of the hepatic artery | 1 (2) | - | - | - |
| None | 1 (2) | - | Operator preference: the RGA was embolized | - |
| Total | 66 (100) | - | *n* = 14 (21%) | *n* = 8 (12%) |

GDA: gastroduodenal artery; RGA: right gastric artery; LHA: left hepatic artery.

A total of 10 of the 66 patients (15%) who received Tc-99m MAA and underwent successful balloon occlusion did not undergo therapeutic $^{90}$Y RE due to the extrahepatic tumor supply in 2 patients, transfer of care in 2 patients, tumor hypovascularity in 2 patients, active medical problems on the day of treatment in 2 patients, and transfer to hospice care of 1 patient. Another three patients underwent therapeutic $^{90}$Y RE without using balloon occlusion due to operator preference for two patients and the development of vasospasm in one patient. Overall, of the initial 67 patients with planned use of balloon occlusion, 53 (79%) successfully underwent therapeutic $^{90}$Y RE, following a diagnostic arteriogram and Tc-99m MAA injection using this technique.

During the study period, an additional 19 patients underwent balloon occlusion for therapeutic $^{90}$Y RE. These patients did not undergo a diagnostic arteriogram or Tc-99m MAA injection with balloon occlusion because they were not planned (*n* = 16) due to repeat treatment (*n* = 2) or due to CHA vasospasm at the time of planning (*n* = 1). Two patients had a prior hepatic resection, consisting of an extended right hepatectomy and left lateral hepatectomy. Overall, the use of balloon occlusion for $^{90}$Y RE administration was successful in all 72 patients. In addition to previously embolized arteries, a total of 12 arteries were embolized during $^{90}$Y RE administration. The purpose of embolization was unsuccessful flow reversal in six arteries (RGA in three, retroportal artery in two, and accessory LGA in one). The remainder of the arteries were embolized for flow redistribution (one LHA and one accessory LHA), operator preference (RGA in two), extrahepatic flow (cystic artery in one), and adjunct treatment (phrenic artery in one). Please refer to Table 3 for details.

Therapeutic $^{90}$Y RE treatment sessions consisted of treatment of the left hepatic lobe only in 4 sessions (with one patient having a prior right hepatectomy), the right hepatic lobe only in 13 sessions, the right hepatic lobe and segment 4 in 1 session, and the whole liver in the remaining 54 sessions. Furthermore, physicians performed dose splitting for 19 of the 59 patients. The mean $\pm$ SD of the administered activity was 47.3 $\pm$ 41.9, 39.4 $\pm$ 21.0, and 62.6 $\pm$ 29.9 mCi for the left lobe, right lobe, and whole liver treatments, respectively. Twenty-two patients received TheraSpheres, and the remaining fifty patients received SIR-Spheres. Complete administration was achieved in 21 of the patients in the TheraSphere group (95%) and 26 of those in the SIR-Sphere group (52%). The reason for the incomplete administration of TheraSpheres in one patient was unknown. However, the reasons for that of the SIR-Spheres were slow flow and stasis in the target vessels in 22 patients, abdominal pain in 2 patients, visualization of an arterioportal fistula during administration in 1 patient, and noncompliance in 1 patient. It is worth noting that target vessel stasis using SIR-Spheres happened during a period when day of calibration spheres and sterile water were being used. Later on, we switched to D5W and a 1-day pre-calibration dose. Post-treatment $^{90}$Y RE scintigraphy confirmed the absence of significant extrahepatic activity in all patients. Complications of therapeutic $^{90}$Y RE are listed in Table 3. The overall complication rate was 10%, and the complications consisted of transient pain during the procedure (*n* = 2), mild spasm at the site of balloon inflation (*n* = 1), minor contrast extravasation during RGA catheterization (*n* = 1), minor arterial dissection (*n* = 1), common femoral pseudoaneurysm (*n* = 1), and a grade 2 duodenal ulcer (*n* = 1). At 90 days post treatment, only one patient had a GI ulcer located in the second portion of the duodenum, and a pathological examination upon biopsy analysis during an upper endoscopy did not show microspheres to support nontarget embolization. This patient eventually recovered with medical management.

**Table 3.** Procedural data and complications following balloon occlusion during therapeutic [90]Y RE.

| Arteries with Flow Reversal | Number of Patients with Flow Reversal (%) | Arteries without Flow Reversal | Coil Embolization during Treatment | Prior Coil Embolization during Planning | Complications |
|---|---|---|---|---|---|
| GDA only | 23 (32) | In one patient, the RGA was not visualized; in three patients, the RGA blood flow did not reverse after balloon inflation; in one patient, both the RGA and retroportal artery flow did not reverse; in one patient, the supraduodenal artery flow did not reverse; in one patient, the retroportal artery flow did not reverse | Flow redistribution: LHA in one patient; operator preference: RGA in two patients; hepatoenteric embolization: RGA and retroportal artery in one patient, retroportal artery in one patient, and RGA in one patients | Flow redistribution: segment IV branch in one patient, middle and left hepatic arteries in one patient *, accessory LHA in two patients, accessory RHA in one patient, and right phrenic artery in one patient; hepatoenteric embolization: RGA in six patients * and accessory left gastric artery in one patient | Microcatheter-induced arterial dissection in one patient |
| GDA and RGA | 31 (43) | A small pancreatic branch from the GDA in one patient and the falciform artery in one patient | Flow redistribution: accessory LHA in one patient; hepatoenteric embolization: RGA in one patient, and accessory LGA in one patient; extrahepatic: cystic artery in one patient | Flow redistribution: replaced RHA in one patient and accessory LHA in two patients; hepatoenteric embolization: accessory LGA in one patient | Duodenal ulcer in one patient, right CFA pseudoaneurysm in one patient managed conservatively, and minor contrast extravasation during RGA catheterization in one patient |
| GDA and retroduodenal artery | 1 (1) | - | - | - | Transient pain during the procedure |
| GDA and supraduodenal artery | 2 (3) | - | - | Hepatoenteric embolization: the RGA was embolized in both patients | - |
| Retroportal artery | 1 (1) | GDA | - | Hepatoenteric embolization: RGA | Transient pain during the procedure before administration |
| GDA, RGA, and supraduodenal artery | 7 (10) | - | The right phrenic artery was bland embolized as adjunct treatment | - | - |
| GDA, RGA, and cystic artery | 1 (1) | - | - | - | - |
| GDA, RGA, and retroduodenal artery | 2 (3) | - | - | - | - |
| GDA, RGA, and segment 4b of the hepatic artery | 1 (1) | - | - | - | - |
| None | 3 (4) | - | - | - | Mild spasm at the site of balloon inflation in one patient |
| Total | 72 (100) | - | *n* = 12 (17%) | *n* = 20 (28%) | *n* = 7 (10%) |

* The same patient had embolization of the RGA and left and middle hepatic arteries. GDA: gastroduodenal artery; RGA: right gastric artery; LHA: left hepatic artery; RHA: right hepatic artery; CFA: common femoral artery.

## 4. Discussion

Since the early days of [90]Y RE, interventional radiologists have creatively used balloon occlusion to achieve various goals. The use of hepatic venous occlusion to perform [90]Y RE in patients with high arteriohepatovenous shunting to reduce the radiation dose to the lungs [11], redistribute the hepatic arterial blood flow in tumors fed by arteries unsuitable for catheterization [12], and protect the hollow viscus from nontarget RE via hepatoenteric blood flow [9,10] are among the applications described in the literature. The balloon occlusion technique used in the present study, the use of a balloon for occluding the CHA, and creating hepatopetal flow in the hepatoenteric arteries were first described by Nakamura et al. [8] in 1985. Physicians used the technique for bland embolization of the whole liver without advancing catheters beyond the CHA. Twelve patients successfully underwent an embolization procedure in that initial experiment. The authors reported side effects related to ischemic cholecystitis but no GI ulcers or pancreatitis following

...

embolization and concluded that this artificially created hepatopetal flow protects the small bowel and pancreas from nontarget embolization.

In comparison with bland embolization, RE has several more layers of complexity. Additionally, nontarget RE to hollow organs may lead to severe complications that may require surgery in a patient population already suffering from multiple comorbid conditions, including cancer. Therefore, interventional radiologists have gone to great lengths to avoid nontarget embolization to the GI tract, including the current practice of coil embolization of the hepatoenteric arteries and skeletonizing the hepatic artery before [90]Y microsphere administration [3]. Even though it has been successful, the current practice comes at a high cost and adds significantly to the operator's and patient's procedure time and radiation dose. The balloon occlusion technique described herein is a modification of the original technique described by Nakamura et al. [8]. As per their description, they used a low-caliber occlusion balloon (i.e., 6 Fr) to temporary occlude the CHA, followed by a careful interrogation of the hepatoenteric arteries and coil embolization of the vessels that did not demonstrate reversed hepatopetal flow. In the 72 patients who underwent [90]Y RE using this technique in the present study, a total of 32 arteries were embolized (<0.5 arteries per patient). It is worth noting is that only 14 arteries (44%) were embolized to prevent nontarget embolization, while the remainder of the embolizations were performed as an adjunct treatment or to redistribute blood flow. Few previous reports have described the occlusion technique in our study. These studies are summarized in Table 4. They include a case report from our center [10] and a report of a phase 1 study by Andrews et al. [9] of 24 patients given TheraSpheres to treat primary or secondary liver tumors. In the latter study, only three patients needed additional embolization of a replaced RHA, replaced LHA, and accessory LHA for blood flow redistribution. The number of vessels embolized per patient was higher in the present study than in the one by Andrews and colleagues, which may be due to the fact that the majority of our patients received SIR-Spheres, which generally includes a higher number of microspheres than TheraSpheres, resulting in a greater chance of nontarget embolization, as well as increased knowledge and awareness of interventional radiologists about the severity of [90]Y RE-related GI ulcers over time.

**Table 4.** Previous reports describing the use of balloon occlusion technique for temporary reversal of flow within hepatoenteric vessels during radioembolization.

| Number | Authors | Year Published | Technique | Patients (Number) | Technical Success (%) | Gastrointestinal Complications |
|--------|---------|----------------|-----------|-------------------|----------------------|-------------------------------|
| 1 | Nakamura, et al. [8] | 1985 | Gelfoam embolization | 12 | 100% | None |
| 2 | Andrews, et al. [9] | 1994 | Glass microspheres | 24 | 100% | 4 (two patients had previously diagnosed gastritis, all biopsy negative for microspheres) |
| 3 | Mahvash, et al. [10] | 2012 | Resin microspheres | 1 | successful | None |
| 4 | Smith, et al. [13] | 2013 | Glass microspheres | 1 | successful | None |

The reported incidence rate for GI toxicity following [90]Y RE is widely variable in the literature, ranging from less than 1% to 45%. Additionally, the GI ulceration rate following [90]Y RE ranged from 0% to 20% (median, 8%), with 6% of cases requiring surgical excision [5]. In one of the largest series of patients undergoing [90]Y RE, consisting of more than 200 patients, the rates of GI toxic effects and ulceration following the treatment were 25% and 12%, respectively [2]. In the present, relatively large, retrospective study, only one patient had an ulcer at 90 days after treatment, and physicians identified no microspheres in the patient's endoscopic biopsy sample. We found that [90]Y RE using the

balloon occlusion technique had a safety profile similar to that of hepatoenteric artery embolization currently used.

This study had some limitations. The first is the potential for sampling bias in patient selection because of the retrospective design of the study. Additionally, the data were obtained at a large tertiary academic center, so the high success and low complication rates in this study may be difficult to reproduce in every center due to differences in interventional radiologist's expertise and techniques. In addition, although this is the largest series to date for which the outcomes of temporary balloon occlusion of the CHA are reported, the study's sample size was still relatively small. Finally, the number and experience level of the operators of the following $^{90}$Y RE and the treatment techniques and methods may have varied slightly over time.

In conclusion, temporary occlusion of the CHA with a balloon catheter is a reliable and reproducible alternative to conventional coil embolization of hepatoenteric arteries during both diagnostic Tc-99m MAA and therapeutic $^{90}$Y RE delivery. This technique's safety profile is comparable to, if not better than, that in the literature regarding GI complications of RE using conventional coil embolization of hepatoenteric arteries.

**Author Contributions:** Conceptualization, A.M.; methodology, A.M.; formal analysis, P.H. and V.G.; investigation, P.H., V.G., B.C.O. and A.M.; writing—original draft preparation, P.H.; writing—review and editing, P.H., B.C.O., V.G., J.D.K., R.A., M.E.A., B.A.C., R.M. and A.M.; supervision, A.M. All authors have read and agreed to the published version of the manuscript.

**Funding:** This research received no external funding.

**Institutional Review Board Statement:** The study was conducted in accordance with the Declaration of Helsinki, and approved by the Institutional Review Board of the University of Texas MD Anderson Cancer Center (protocol code PA12-0425 and date of approval 09/04/2012).

**Informed Consent Statement:** Patient consent was waived by the Institutional Review Board of the University of Texas MD Anderson Cancer Center due to retrospective nature of the study.

**Data Availability Statement:** The data presented in this study are available on request from the corresponding author. The data are not publicly available due to lack of Institutional Review Board approval for public sharing.

**Acknowledgments:** The authors would like to thank Monroe Donavan Anderson Medical Library and Donald R. Norwood for kindly editing the final version of the manuscript.

**Conflicts of Interest:** The authors declare no conflict of interest.

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
