# Peer review of "Temporary Reversal of Hepatoenteric Collaterals during 90Y Radioembolization Planning and Administration"

_curroncol, doi:10.3390/curroncol29120753_

Round 1
Reviewer 1 Report
This paper documents the results of Experienced Interventional Radiologists using reversible balloon occlusion instead of irreversible thromboembolization to exclude small arterial branches that would allow the 90Y to go to and damage soft tissue. I feel it is a worthwhile paper to be published and is of particular interest to Hepatobiliary-pancreatic and Liver Transplant surgeons as well as to Interventional Radiologists at Hospitals treating patients with primary and metastatic liver cancers. However, I found it exceedingly difficult to understand what the authors were discussing at times, because I felt they were using terminology that is possibly unique to their specialty but not clear to physicians outside of interventional radiology. So I would like them to change their language to make it clearer to those of us who send them patients, as this paper is to be published in a non-radiology journal. I will try to give examples and hopefully the authors can go through the paper and make appropriate changes throughout. I also suggest other editing and verbiage changes.
1.) In line 20 of the abstract, you wrote "temporarily redirect blood flow from the hepatoenteric arteries". I spent awhile trying to understand what you meant by "redirect". Actually, you are not redirecting the blood flow, which mean placing a bypass; what you mean is that you blocked the blood flow or excluded the blood flow by occluding the artery temporarily with the balloon.
2.) To what are you referring by writing "both radionuclides" in line 21 in the Abstract? Up until then, you have mentioned only one radionuclide.
3.) In line 22, you wrote "11 patients required embolization due to unsuccessful flow reversal with (the) balloon occlusion technique". I disagree that the balloon is causing flow reversal, it is simply blocking the blood flow. However, if you are assessing that the blood flow is blocked by observing that the arterial flow in the collaterals is going in the opposite direction, then if makes sense but you need to spell that out, by saying something like, "When we inflated the occlusion balloon and saw reversal of the flow in the collaterals we were trying to protect, then we felt that the balloon occlusion was effective."
Author Response
Thanks to reviewer 1 for kindly reviewing our manuscript. We have made modifications based on the comments. 1.) In line 20 of the abstract, you wrote "temporarily redirect blood flow from the hepatoenteric arteries". I spent awhile trying to understand what you meant by "redirect". Actually, you are not redirecting the blood flow, which mean placing a bypass; what you mean is that you blocked the blood flow or excluded the blood flow by occluding the artery temporarily with the balloon.
Answer: We modified the sentence to "...catheter in the CHA to temporarily direct blood away flow from the hepatoenteric arteries were..."
2.) To what are you referring by writing "both radionuclides" in line 21 in the Abstract? Up until then, you have mentioned only one radionuclide.
Answer: We rephrased the sentence as below to reflect this: "SPECT/CT nuclear scintigraphy was performed after both diagnostic and treatment angiogramsdelivery of both radionuclides"
3.) In line 22, you wrote "11 patients required embolization due to unsuccessful flow reversal with (the) balloon occlusion technique". I disagree that the balloon is causing flow reversal, it is simply blocking the blood flow. However, if you are assessing that the blood flow is blocked by observing that the arterial flow in the collaterals is going in the opposite direction, then if makes sense but you need to spell that out, by saying something like, "When we inflated the occlusion balloon and saw reversal of the flow in the collaterals we were trying to protect, then we felt that the balloon occlusion was effective."
Answer: Thanks to reviewer 1, we re-wrote this sentence to reflect this comment as below: "Overall, only 12 hepatoenteric arteries in 11 patients required embolization due to persistent unsuccessful flow reversal persistent hepatoenteric flow with despite balloon occlusion technique"
Reviewer 2 Report
Interesting study. I have the following comments:
1) The term radioembolization instead of the non-standard term radioembolotherapy should be used.
2) The bibliography is too poor and some relevant references on the topic of radioembolization should be added, such as PMID: 27579537
3) I would add a table on the previous reports of this event in the literature
4) Do the authors think that combination of TARE with antiangiogenic therapy could lead to reversal of these collaterals? (see the paper PMID: 32272656 )
Author Response
1) The term radioembolization instead of the non-standard term radioembolotherapy should be used.
Answer: Thanks to reviewer 2. We revised the manuscript to reflect this.
2) The bibliography is too poor and some relevant references on the topic of radioembolization should be added, such as PMID: 27579537
Answer: The articles on this specific technical subject (balloon occlusion for flow reversal) are very limited and only a few reports with small number of patients exists reference 7-9. There is overwhelming evidence on the rule of radioembolization in different liver primary or secondary malignancies. However, these are not directly related to this subject.
3) I would add a table on the previous reports of this event in the literature
Answer: Thanks to reviewer 2. We actually entertained the idea and discussed it but there are only 2-3 small reports on this specific subject and we already had a few tables so we decided to only cite the previous reports in the manuscript.
4) Do the authors think that combination of TARE with antiangiogenic therapy could lead to reversal of these collaterals? (see the paper PMID: 32272656 )
Answer: Thanks to reviewr2 and we believe this is a great question. These hepatoenteric collaterals are usually medium sized vessels that could be seen on pre-procedure CT. We have been treating a number of HCC patients on anti-angiogenic medications and we still see these collaterals in this patient population. We believe an independent study comparing patients treated on these medications versus the rest of the patients would be suitable to answer this question.
Round 2
Reviewer 2 Report
Ok, the manuscript is fine in the current form. It can be accepted.